# Nutrition of Infants with Bronchopulmonary Dysplasia before and after Discharge from the Neonatal Intensive Care Unit

**DOI:** 10.3390/nu14163311

**Published:** 2022-08-12

**Authors:** Ageliki A. Karatza, Despoina Gkentzi, Anastasia Varvarigou

**Affiliations:** Department of Paediatrics, Neonatal Intensive Care Unit, University of Patras Medical School, 26504 Patras, Greece

**Keywords:** extremely preterm neonate, extremely low birth weight infant, bronchopulmonary dysplasia, nutrition, feeding

## Abstract

Bronchopulmonary dysplasia (BPD) represents a severe sequela in neonates born very prematurely. The provision of adequate nutritional support in this high-risk population is challenging. The development of the lungs and physical growth are closely linked together in infants with BPD. Growth deficiency has been associated with pulmonary dysfunction, whereas improvement in respiratory status results in growth acceleration. Currently, there is not enough data regarding optimal nutritional strategies in this population. Nutrition in these infants should provide sufficient calories and nutrients to establish growth, avoid growth retardation and assist alveolarization of the lungs. Meticulous follow-up is mandatory during and after discharge from the Neonatal Intensive care Unit (NICU) to minimize growth retardation and improve lung function. Despite the significant literature supporting the contribution of growth and nutrition in the avoidance of BPD, there is limited research regarding interventions and management of infants with established BPD. Our aim was to review clinical strategies applied in everyday clinical practice and identify debates on the nutritional approach of newborns with BPD. Well-organized interventions and clinical trials regarding the somatic development and nutrition of infants with BPD are warranted.

## 1. Introduction

Bronchopulmonary dysplasia (BPD) represents a severe sequela in neonates born very prematurely [1]. Bronchopulmonary dysplasia was defined in 1967 by Northway as a disorder of the respiratory system in late preterm newborns who were supported for a long time with artificial ventilation and required increased administration of oxygen [2]. In the “Old BPD” cystic changes of the lungs and heterogeneous aeration have been documented, whereas the “New BPD” has a different phenotype and presents in extremely preterm neonates who have been exposed to antenatal steroids, have been given surfactant and have been supported with gentle modern ventilator techniques [3]. The main feature of new BPD is diffuse underdevelopment of the alveoli, which leads to a significantly diminished lung substrate for the elimination of CO_2_ and the accretion of O_2_, injury of the airways, inflammation, and pulmonary fibrosis. The alterations in the lungs are less profound than in old BPD [4]. BPD affects predominantly premature neonates born at a gestational age which is less than 28 weeks with 67% of them affected by a moderate or severe form of BPD resulting in a mortality of 19% [5]. The incidence of BPD remains stable despite the great progress in perinatal care leading to improved survival of very premature neonates [1].

BPD takes place in neonates born at the developmental stage of the lungs in which the transition from the canalicular to the saccular stage occurs. Its pathophysiology is multifactorial involving several antenatal and postnatal adverse events which derange lung development leading to severe disease with long-term consequences [6]. Chronic deprivation of nutrients and oxygen in pregnancy may result in alterations in the structure of the airways and the lung parenchyma affecting lung function, not only in neonates but also later in life. Deficient nutritional support in very premature infants deranges the development of the respiratory system and may contribute to the evolution of BPD [7]. Prematurity affects lower respiratory tract embryogenesis which has been attributed to unfavorable antenatal and postnatal events such as intrauterine growth retardation, inflammation, resuscitation, oxygen demand, ventilatory support and systemic infections, all of which can result in an arrest in pulmonary vascular development and alveolarization [3]. It is not known whether prematurity causes BPD or whether parameters that are associated with prematurity are its cause [6]. In children with severe BPD, the performance of the respiratory system does not ameliorate with chronological age. In a recent study, in two-thirds of the subjects enrolled, Forced expiratory volume at 1 s (FEV1) and the ratio to forced vital capacity (FEV_1_/FVC) got worse over time. Lung function progress was better in adulthood in the subjects that were born after 1990 in comparison with children born before this period. This may be attributed to improved perinatal care in recent years [8]. Pulmonary hypertension has increased incidence in children with BPD. It is documented in 25% of children in the severe spectrum of BPD and represents a parameter that adversely affects morbidity and mortality in this population [9]. Survivors with neonatal BPD also have neurological sequelae, affecting motor and cognitive development and have lower academic progress compared to premature infants without BPD [10].

The real incidence of BPD is difficult to identify as different definitions have been used. This problem makes also difficult to identify variations in the incidence of BPD over time [11,12]. In a study performed in the USA during the last 20 years, improvements in pregnancy and neonatal management and minimization in overall morbidity have been documented; however, the incidence of BPD increased. Increased survival was documented for infants who were born at 23 to 24 weeks and survival without significant sequelae was shown for neonates that were born between 25 and 28 weeks [13]. BPD affects predominately neonates whose birth weight is less than 1500 g. There is an inverse relationship between the prevalence of BPD and gestational age as well as birth weight [13]. In Europe, 10–20% of infants with a gestational age of 23–31 weeks manifest BPD [14].

Increased fluid provision and inefficient loss of weight occurring in the first 10 days after birth have been linked to the augmented likelihood of BPD because of pulmonary congestion and edema [15]. Furthermore, it has been shown that in neonates born at less than 30 weeks, sodium provision should be deferred after 6% of birth weight is lost, an intervention that decreases oxygen requirements without affecting growth [16]. To avoid the development of BPD, extremely preterm and very preterm infants are often fluid-restricted to 120 mL/kg/day [12,15]. Commonly diuretics are prescribed; however, current information on the effectiveness of diuretics in the avoidance and therapy of BPD has been questionable [17]. Fluid restriction can lead to deficient caloric provision, compromised growth, and inadequate nutritional support, which are risk factors for BPD development [12,18]. It should be noted that premature neonates with established BPD have higher demands in calories because their work of breathing is higher and they require caloric support to assist in the alveolarization of the lungs and overcome pulmonary inflammation [19,20]. Preterm infants who develop BPD need medical support and supervision beyond neonatal intensive care unit hospitalization, including nutritional support and monitoring of growth status in the long term [6].

## 2. Growth of Neonates with Bronchopulmonary Dysplasia

The nutritional support of very preterm newborns was investigated in a cohort study that reported that 20.5% developed BPD. At term equivalent anthropometric measurements were smaller in the bronchopulmonary dysplasia group who initiated feeding by mouth and achieved full oral feeds at a more mature postnatal age, requiring prolonged intravenous nutritional support. The amount of protein given to both groups was the same, irrespective of whether they had bronchopulmonary dysplasia or not, but fluid volume and proteins given to those who developed BPD were lower after the fourteenth day. The ratio of calorie to protein (30 kcal/g protein) per day was achieved by almost 90% of the neonates that had no bronchopulmonary dysplasia on their 21st day of life. The bronchopulmonary dysplasia group received a lower amount until the fourteenth day and the deficiency persisted in 56.3% of them for one month. The authors conclude that an inadequate calorie/protein ratio was a fact in neonates who established bronchopulmonary dysplasia, and adequate nutritional support for this population is challenging. Somatic parameters were measured at birth and at term equivalent postnatal age apart from the weight which was measured every week for four weeks. Z-scores at birth were similar between the neonates who developed BPD and those who did not develop BPD. However, at term equivalent, head circumference and length Z-scores were significantly lower in infants with BPD. The decrease in the weight Z-scores in both groups was similar during the first month, but at term equivalent to postnatal age, the Z-scores of the weight of the premature neonates with BPD showed further decrease, while the preterm infants without BPD started to catch up [21].

Another study investigated the nutritional condition of neonates with BPD during the first month after birth and the probability of acquiring malnutrition. The infants in the BPD group received excessive fluids but fewer calories and proteins after the first 14 days of life, required mechanical ventilatory support for more days and needed more days to achieve oral feeding without the need for supplementation with parenteral nutrition, which was thus administered for more days. These disorders were more prevalent in the group of infants with malnutrition. Anthropometric measurements were significantly smaller in the BPD in comparison to the non-BPD infants and they were lower in infants with malnutrition than in those who were well nourished. The weight to length ratio, body mass index, triponderal mass index, and the velocity of weight gain in the population with BPD was significantly lower compared to the non-BPD group.

Days on artificial ventilatory support, provision of lipids at 4 weeks, time to achieve full feeding by mouth and parenteral nutrition days were independent risk factors for malnutrition [22].

Malikiwi et al. have investigated if there is a link between growth status, provision of nutrients and fluids in the first month after birth and the probability to acquire bronchopulmonary dysplasia. Infants with BPD received fewer calories and fluids during the first month of life. Z-scores of weight at birth and at one month were similar between the BPD and non-BPD neonates. The percentage of infants with weights below the 10th percentile and the mean weight velocity at one month also were similar between the groups. However, the adverse effect of BPD on somatic growth was preserved during infancy, so infants with BPD had significant growth retardation at 6 and 12 months of age.

The risk of establishing BPD was higher when infants were ventilator-dependent during the first month and were lower when a higher first-month provision of calories was achieved. The authors conclude that the BPD group received fewer calories and fewer fluids during the first month. Only two independent predictors for BPD establishment were identified and these were the dependency on an artificial ventilator and the provision of low caloric intake during the first month [23].

Another study investigated whether there is a relationship between the provision of nutrients during the first week of life and the severity of bronchopulmonary dysplasia in neonates born extremely preterm. Out of the 226 infants that were enrolled in the study 67% developed moderate–severe BPD. Participants with moderate–severe BPD were more immature, had a lower birth weight, and needed more prolonged ventilatory support compared to the infants that did not develop BPD. During the study period, the provision of nutrients was significantly less, whereas the provision of fluids was significantly more in the group who developed moderate–severe BPD compared to those who did not. Taking into consideration potential confounding factors, fluid provision, time on mechanical ventilation and low caloric provision were significant determinants of increased risk of moderate–severe BPD [24].

Another study aimed to investigate the role of nutritional support either by mouth or intravenously during the first 2 weeks of life to the risk of acquiring bronchopulmonary dysplasia. The neonates who acquired BPD received fewer calories compared to the controls. Carbohydrate and fat provision was also diminished in BPD patients. This study showed that neonates with BPD are provided with fewer calories and lipids during the first two weeks of life [25].

Preterm infants with BPD have growth retardation, with deficient total body fat and free fat mass identified at 6 weeks post-term; free fat mass and total body fat remain low compared with healthy term infants during the first year of life [26].

As already stated, premature neonates with BPD have an increased incidence of restriction in growth after birth. In a multicentered cohort study breastmilk feeding used exclusively was related to diminished growth but a reduction in the probability of developing bronchopulmonary dysplasia, Furthermore, the incidence of necrotizing enterocolitis and retinopathy of prematurity were decreased [27]. In a retrospective case–control study, neonates with bronchopulmonary dysplasia were provided with fewer calories and fluids in the first month of life. After adjustment for confounders, the requirement for mechanical ventilation and low one-month provision of calories were independently related to increased risk of BPD [23]. As previously mentioned energy requirements are increased in infants with BPD [19,20]. Another study evaluated whether providing more oral calories in preterm neonates with BPD can have a positive effect on growth. Growth was assessed in a cohort of premature neonates who were born before 32 weeks of gestation, their birth weight was less than 1500 g and needed oxygen support up to 28 days after birth who were fed individually tailored fortified breast milk and/or preterm formula, and was compared to a group of neonates with BPD who received fortified breast milk and/or pre-term formula [28]. Days of parenteral nutrition and oral protein intake did not differ between the two groups. However, the provision of oral calories was increased in the infants in the first group with mild or moderate BPD and in those with severe BPD. Rates of weight increase were higher in the neonates of the first group with mild or moderate BPD and in those with severe BPD. The percentage of neonates with growth restriction at 36 weeks was increased in the second group. The authors suggest that optimization of nutritional support improves growth rates after birth in premature neonates having bronchopulmonary dysplasia [25,28,29].

## 3. Nutritional Management in Infants with Established BPD, either in the Hospital or after Discharge

Currently, few data describe the ideal nutrition strategy in neonates with BPD before and after the exit from the NICU [30]. A recent meta-analysis with the use of PRISMA guidelines with thirty articles selected for inclusion concluded that although there is a lot of data supporting the significance of growth and nutritional support in the avoidance of BPD, there is limited research regarding policies and treatment of infants with BPD. Therefore, interventions and well-organized trials concerning growth and nutritional support in subjects with BPD are warranted. These ideally have to be multicenter as cases with BPD are few at each NICU [30].

Lung function and physical growth are interrelated in neonates developing bronchopulmonary dysplasia. Growth retardation is related to prolonged respiratory impairment and conversely, amelioration in lung function results in increased growth rates [31].

Energy demands in infants with BPD are 15–20% higher compared with infants without BPD [19,20]. This is probably due to laborious breathing, tachypnoea and oxygen demands. Patients born extremely preterm may have difficulties with oral feeding and experience gastroesophageal reflux, vomiting and other issues associated with discoordinated sucking, swallowing dysfunction, poor swallow breath coordination, and poor sucking endurance and performance [19,20]. Early recognition of feeding difficulties and gastroesophageal reflux is crucial for the nutritional management of these infants. Chronic stress and inflammation as well as the use of diuretics and corticosteroids may increase energy demands. Furthermore, necrotizing enterocolitis may require bowel resection and may result in short bowel syndrome with resultant malabsorption and limited energy intake orally or via intravenous nutritional support. Total parenteral nutrition is essential at the initial age of all very premature infants but may be warranted for the treatment of patients with short bowel syndrome in the long term [32].

Fluid restriction is essential for infants with BPD who should receive no more than 150 mL/kg/day and ideally 135 mL/kg/day [33,34,35]. The oral provision of energy should be optimized at 120–150 kcal/kg/day [30,34]. However, it is not always easy to establish satisfactory caloric provision by offering a restricted volume of fluids. A limited number of studies have investigated the needs of neonates with BPD in proteins, therefore it may be suggested that these are similar to the requirements of premature neonates that do not have BPD according to ESPGHAN [36]. These subjects have altered body composition, suggesting that the usual provision of protein may not be adequate [20]. Enteral protein intake should be at 4.0–4.5 g/kg/day in infants having a birth weight of less than 1000 g and at 3.5–4.0 g/kg/day in neonates with a birth weight of 1000–1800 g [36].

Lipids are an essential component for the growth of extremely premature neonates because they include essential fatty acids and assist to achieve energy requirements [36]. Moreover, lipids play an important role in the absorption of fat-soluble vitamins [20]. Total lipid intake should be 4.8–6.6 g/kg/day with 12–30 mg/kg/day of Arachidonic acid and 18–42 mg/kg/day of Docosahexaenoic acid [36]. Neonates should be provided with long-chain polyunsaturated fatty acids (LC-PUFAs) in adequate amounts to assist visual and cognitive development. For breastfeeding infants, LC-PUFA is provided by maternal milk; however, when breastfeeding is not the case, dietary LC-PUFAs should be supplemented during the first six months of life. Currently, there is not sufficient information for quantitative recommendations regarding LCPUFAs provision [37]. The results of a very recent meta-analysis were that provision with n-3 polyunsaturated fatty acids cannot prevent BPD. Therefore, the increased use of n-3 polyunsaturated fatty acids in premature infants to prevent BPD is not currently supported [37].

The recommended nutritional intakes for premature infants with bronchopulmonary dysplasia are presented in the table.

A bone disease of prematurity is an issue in neonates with BPD and has been associated with the use of postnatal steroids, diuretics and deficient intake of minerals [38].

Enteral feeding should provide a sufficient quantity of calcium and phosphorus; however, it is usually deficient due to its low amount in oral feeds, which has been attributed to the usage of breast milk without fortification, cholestatic liver disease or disorders of absorption [39]. Moreover, bone mineral deposition is reduced in infants with BPD who receive corticosteroids and diuretics usage increases the excretion of calcium. As expected osteopenia of prematurity, a disorder that is due to the insufficient nutritional provision of calcium and phosphorus is an important issue in neonates with bronchopulmonary dysplasia [34,35,37]. Enteral calcium and phosphorus intake should be optimized at 120–140 mg/kg/day of Ca or 150–220 mg/kg/day with 90 mg/kg/day or 75–140 mg/kg/day of phosphorus and Ca/P ratio of 2 [38,39]. In order to avoid volume overload diuretics are often prescribed causing hyponatremia as a complication that requires the provision of sodium which is necessary for optimal growth. Ideally serum Na levels should be >135 mEq/L [40]. Vitamin A and Vitamin E should be supplied at 400–1000 µg/kg/day or 1320–3300 IU/kg/day and 2.2–11 mg/kg/day, respectively [40]. The role of avoidance and therapy of anemia is also important and therefore iron supplementation is required at 4 mg/kg/day, starting at 1–2 months after birth and up to the age of 12 months [41].

Preterm infants have low selenium levels as its transfer through the placenta occurs mainly during the third trimester of pregnancy. The American Society for Clinical Nutrition (ASCN) recommends a parenteral Se intake of 2 μg/kg/day and the American Academy of Pediatrics Committee on Nutrition recommends enteral Se provision of 1.3–4.5 μg/kg/day in preterm neonates [42]. Darlow et al. found an association between low plasma selenium levels and increased risk of lung disease in very premature infants, defined as oxygen dependency on the 28th day of life [43]. A Cochrane review from 2003 showed that low plasma Se was associated with increased complications of prematurity including BPD, increased days of oxygen dependency, and increased risk of adverse respiratory outcomes [44].

Zinc promotes epithelial development, participates in the enzymatic reactions underlying the repair of tissue damage, protects against infection, and modulates the inflammatory response of the respiratory system [45]. Thus, it is reasonable to hypothesize a potential contribution of zinc in preventing bronchopulmonary dysplasia. However, clinical studies demonstrating a clear relationship between zinc and BPD are not yet available [46].

## 4. Human Breast Milk

A recent meta-analysis ιinvestigated the role of donor milk on the incidence of BPD in comparison with feeding with formula and found that the incidence of BPD was significantly lower in the donor milk-fed group [47]. Similar were the results of another study that compared feeding exclusively fresh maternal breast milk compared to maternal breast milk after pasteurization [48]. The participating infants were divided into two groups: one group received their mother’s own fresh milk and another group was given their mother’s milk post pasteurization. After controlling for confounding factors, the incidence of bronchopulmonary dysplasia was lower in the group who received their own mother’s fresh milk. The provision of human milk exclusively, preferably using fresh maternal breast milk, is recommended in the treatment of neonates at the early stages of BPD [49]. The results of a large study involving premature neonates born before 32 weeks showed that maternal breastmilk used exclusively resulted in a decreased incidence of BPD, necrotizing enterocolitis and retinopathy of prematurity, although it was associated with decreased growth [50]. Furthermore, in infants with established BPD provision of breast milk for a longer period of time was related to reduced visits to the hospital emergency departments, less corticosteroid use, cough or chest congestion, and a lower incidence of admissions to the hospital [49].

Maternal milk is generally accepted as the ideal feeding for infants in the first 6 months after birth [51]. Breast milk has been proven to have short and long-term benefits and can be considered a public health issue. Hospitals should routinely support and encourage the initiation and continuation of breastfeeding exclusively during the first 6 months of life as suggested by the American Academy of Pediatrics and WHO/UNICEF [51].

In oxygen-dependent infants with BPD, a link has been documented between the time of oxygen support and gestational age, duration of ventilator support, hemoglobin concentration at the exit from the NICU, use of human milk exclusively and growth. An ambulatory Kangaroo Mother Care Program with established protocols and close follow-up showed that breastfeeding permits sufficient weight gain and assists stop of oxygen use in a safe way. Anemia, growth, and oxygen weaning in oxygen-dependent premature infants were significantly related to breastfeeding in this program [52].

Although breastfeeding should be supported and current evidence indicates its use as the ideal form of feeding for all neonates has positive consequences for human health both in infancy and childhood, human milk may not offer adequate nutritional support for very premature neonates. Using volumes of 120–150 mL/Kg may lead to impaired growth rates resulting in poor health outcomes including bronchopulmonary dysplasia. Fortification of maternal milk with a commercial product containing protein, calcium, and phosphate is suggested to achieve the higher nutritional needs of the very low birth weight neonates [53].

Although fortified maternal milk is the preferred type of feeding, nutrients provided by fortified breast milk often are insufficient to cover the nutritional requirements of neonates with bronchopulmonary dysplasia. This can be attributed to the low protein concentration of the fortifiers and changes in human milk’s nutritional values over time. Individualized human milk fortification has been suggested, such as adjusted fortification and target fortification to overcome this issue [20].

Adjustable fortification consisted of standard fortification and the addition of more quantity of fortifier and extra protein monitored by blood urea nitrogen measurements. Infants receiving the adjustable fortification regimen had increased provision of proteins during the first 3 weeks after birth and showed significantly greater anthropometric measurements compared to neonates in which the standard fortification formula was used. Anthropometric measurements were positively related to protein provision [54].

In another interventional cohort study, BPD infants were fed with specifically fortified breast milk and/or preterm formula with the addition of Duocal and MCT oil and were compared with a control group fed fortified breast milk and/or pre-term formula. The intervention group showed improved growth suggesting that optimization of nutritional support improves growth after birth in preterm infants with BPD [55]. Although the increased provision of calories may result in increased CO_2_ release, the favorable growth outcome is more important in infants with BPD [20].

## 5. Premature Formulas and Caloric Supplementation

A study that enrolled 224 infants with BPD found that almost half started premature formula before exiting the NICU. Neonates who stopped human milk before exiting the NICU received human milk for a shorter period of time compared with those who continued to breastfeed after discharge [55]. Following the cessation of maternal milk based on the maturity of the infant and nutritional demands, feeding with a specific formula is suggested. The commercially available preterm formulas which are used in the NICU, contain increased amounts of energy, protein, calcium, and phosphorus, and the type of fat added in the formula is a blend of vegetable oils including long-chain triglycerides and MCT [56].

A nutrient-enriched formula containing high energy and micronutrients used on neonates with BPD up to 3 months after birth results in improved growth in comparison to neonates consuming an isoenergetic standard preterm formula. This implies that infants with BPD have increased growth rates when formulas richer in nutrients compared to standard ones are used [57].

Theile et al. have demonstrated that neonates with bronchopulmonary dysplasia born in the current period have better growth compared with infants born 10 years ago. The authors speculate that early parenteral nutrition including amino acids and caloric-dense feeding policies have contributed to this result. They speculated that growth acceleration depends on several factors; however, the administration of calorically dense (>24 kcal per oz) milk products that optimize the uptake of proteins in infants in which liberal provision of fluids is contraindicated plays an important role. Calorically dense milk contains about 3 g of protein/100 kcal compared to formulas given ten years ago, which provided about 2 g of protein/100 kcal. BPD patients in the current era BPD had improved growth and required ventilator support for a shorter period of time [58].

A pilot study compared the nutritional status of patients with BPD who were provided with either ready-to-feed formula having a caloric concentration of a 30 kcal/oz or a preterm formula including nutritional supplementation. The authors found that the nutritional supply of the two regimens was similar, whereas the 30 kcal/oz formula provided increased protein. Thus the 30 kcal/oz formula is a satisfactory choice for premature infants with BPD [59].

Following cessation of breast milk after discharge from the NICU post-discharge formulas improve growth rates. There is no sufficient information supporting the use of certain formulas in neonates with bronchopulmonary dysplasia. As respiratory function improves in parallel with growth velocity high caloric milk can be weaned and children can follow a normal diet [20]. However, a percentage of children with BPD which is severe will need high caloric supplementations in their childhood to maintain sufficient growth [20]. In addition, the introduction of solid foods may be different in term infants compared to those with BPD. The initiation of solid foods should not be based on chronological age as these children may be able to swallow solid foods at a later age as a result of prematurity and feeding difficulties. Thicker foods may easier to swallow [19].

## 6. Feeding Issues

Gastroesophageal reflux is a frequent problem in neonates that have bronchopulmonary dysplasia. A study of 131 infants with BPD assessed the occurrence of side effects with a follow-up time of 1.5 years. pH-multichannel intraluminal impedance and gastric sodium concentrations were assessed for 24 h in the study population when they reached 36 weeks and 18 months. The incidence of gastroesophageal reflux in BPD was about 40% and included both acid gastroesophageal reflux and duodenogastroesophageal reflux. Increased incidence of respiratory symptoms was documented in infants born before 30 weeks, having a birth weight less than 1500 g, those requiring mechanical ventilation > 7 days, and those who had had acid and duodenal gastroesophageal reflux. Infants with BPD and duodenal gastroesophageal reflux were at higher risk for late side effects compared to the rest of the infants studied [60].

Swallowing dysfunction is also a common problem in BPD patients that may affect respiratory function. Neonates that develop bronchopulmonary dysplasia have tachypnoea, experience more episodes of low saturations, have poor coordination between sucking and swallowing and have poor sucking endurance and performance. They also have recurrent episodes of cough, wheezing, vomiting, difficulties with feeding and choking [61].

## 7. Alternatives to Oral Feeding

As infants with BPD have a high incidence of difficulties with oral feeding very often alternative methods of feeding are applied, even after discharge home. Feeding by mouth has been shown to affect respiratory function in very premature neonates with severe BPD during NICU stay or even when they go home. This is a prolonged effect as these infants experience significantly lower saturations during feedings even at 2–6 months after birth. Another side effect documented in this population is that they also have growth restrictions at the same chronological age [61].

Studies concerning feeding strategies after discharge from the NICU have demonstrated that the use of nasogastric tube feeding is preferable to gastrostomy [62]. However, different institutions use different feeding protocols post-discharge. Nasogastric tube use in the population with established bronchopulmonary dysplasia seems to be a practical approach with lower numbers of babies needing the insertion of a gastrostomy tube. Brain magnetic resonance imaging abnormalities were more prevalent in infants in babies in which a gastrostomy tube was placed postdischarge [63].

Analysis of very premature infants from 25 centers found that among the infants requiring gastrostomy tube placement post-discharge, 77% had bronchopulmonary dysplasia. The gastrostomy tube was related to altered anthropometric parameters, neurodevelopmental delay, and feeding and prolonged lung issues at follow-up [64].

## 8. Monitoring of Growth and Nutritional Status

While being in hospital weight is monitored every day and length and head circumference once a week. Iron status should be monitored with complete blood count with reticulocyte count, serum ferritin levels, protein status with blood urea nitrogen and metabolic bone disease with phosphorus and alkaline phosphate concentrations. Regular electrolyte measurements should be measured in patients on diuretics and monitoring of vitamins and trace elements should be performed if there is suspicion of deficiency [20]. A similar plan should be employed after discharge with the general pediatrician and the aid of specialists in nutrition and feeding. The implementation of a multidisciplinary approach involving the consultation of specialists may be beneficial for these infants.

## 9. Conclusions

Nutrition of infants with BPD should focus on the supply of adequate caloric support and provision of nutrients to promote growth, prevent extrauterine growth retardation and assist in alveolarization of the lungs. Meticulous follow-up is warranted before and after discharge from the NICU to minimize growth deficiency and improve lung function. Currently, there is limited data regarding the optimal feeding strategies for this population of infants. Well-designed clinical studies are required to improve our practices regarding feeding and nutrition issues of infants with BPD.

## Data Availability

N/A.

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
