# Peer review of "Nutrition of Infants with Bronchopulmonary Dysplasia before and after Discharge from the Neonatal Intensive Care Unit"

_nutrients, 2022, doi:10.3390/nu14163311_

Round 1

Reviewer 1 Report

The manuscript by Karatza et al reviewed the current nutritional approaches in the infants with bronchopulmonary dysplasia (BPD). The paper discussed growth of neonates with BPD, nutritional management in infant with BPD, importance of breast milk in BPD and provided insight on the alternative to oral feeding. This review paper is well written and will provide useful information about nutritional approaches in infants with BPD to the readers. I do have following comments and suggestions that might help to improve the quality of the manuscript.

Page 2; Please define FEV, FVC when first mentioned.

Previous studies have suggested the importance of selenium for preterm infants. I would like to suggest authors to discuss the selenium and other trace element’s role in BPD.

Authors have mentioned the recommended intake for AA and DHA. Is there information about other polyunsaturated fatty acids (PUFAs)? 

I would strongly suggest including table or schematic illustration that demonstrate the nutritional approaches in infants with BPD.

Author Response

Reviewer 1

Dear sir/madam,

thank you very much for the very interesting comments. Your questions have been addressed one-to-one and corrections appear in red in the corrected manuscript

Page 2; Please define FEV, FVC when first mentioned.

The words have been changed by Forced expiratory volume at 1 second (FEV1) and forced vital capacity (FVC)

Previous studies have suggested the importance of selenium for preterm infants. I would like to suggest authors to discuss the selenium and other trace element’s role in BPD.

The role of selenium and Zinc in BPD have been discussed and the appropriate references added

Authors have mentioned the recommended intake for AA and DHA. Is there information about other polyunsaturated fatty acids (PUFAs)?

Currently there is not sufficient information for quantitative recommendations regarding LCPUFA provision 

I would strongly suggest including table or schematic illustration that demonstrate the nutritional approaches in infants with BPD.

A table demonstrating the nutritional support of infants with BPD has been added

Reviewer 2 Report

The authors present an interesting review on the nutrition of infants with bronchopulmonary dysplasia. Their analysis covers the nutritional support during the first weeks (calories and fluid), use of human milk, premature formula, feeding issues, the use of nasogastric tubes or gastrostomy, and the monitoring of the growth and nutritional status of infants. 

The paper seems to be well written but, still, I think that there could be some improvements:

- the title may be improved by letting out the "before and after discharge..."

- The first chapter 1. Growth of neonates with BPD may be better organized as it contains repeated data coming from different studies. There are few data on infants' growth, and most of the data are on the nutritional supply.

- I think that it would be better to have the main recommendations covered by this review collected into a table; to have some practical guidelines for the readers.

- there are a few minor editing issues: use BPD all the time after the first definition; explain other abbreviated words when used first (NICU).

Author Response

Reviewer 2

Dear sir/madam,

thank you very much for the very interesting comments. Your questions have been addressed one-to-one and corrections appear in red in the corrected manuscript

- the title may be improved by letting out the "before and after discharge..."

The words have been deleted from the title

The first chapter 1. Growth of neonates with BPD may be better organized as it contains repeated data coming from different studies. There are few data on infants' growth, and most of the data are on the nutritional supply.

The first chapter was better organized containing information on infants’ growth as indicated by the papers included in the chapter. Also an extra reference was added offering data regarding growth of infants with BPD

- I think that it would be better to have the main recommendations covered by this review collected into a table; to have some practical guidelines for the readers.

The main recommendations for nutritional support have been collected into a table to offer some practical guidelines to the readers

- there are a few minor editing issues: use BPD all the time after the first definition; explain other abbreviated words when used first (NICU).

BPD was used at all times after first definition and NICU was explained when used for the first time